# Activity–Rest Circadian Rhythm of the Pearly Razorfish in Its Natural Habitat, before and during Its Mating

**DOI:** 10.3390/biology12060810

**Published:** 2023-06-02

**Authors:** Mourad Akaarir, José Manuel Pujol, Margalida Suau, Rubén V. Rial, María Cristina Nicolau, Antoni Gamundi, Martina Martorell-Barceló, Margarida Barceló-Serra, Eneko Aspillaga, Josep Alós

**Affiliations:** 1Laboratorio del Sueño y Rítmos Biológicos, Universitat de les Illes Balears, IDISBA, IUNICS, 07122 Palma, Spain; 2Fish Ecology Group, The Mediterranean Institute for Advanced Studies, IMEDEA (CSIC-UIB), 07190 Esporles, Spain

**Keywords:** high-resolution acoustic tracking, free-living marine fish, activity-rest circadian rhythm, pearly razorfish

## Abstract

**Simple Summary:**

Understanding the circadian rhythm of activity–rest in marine fish is essential for comprehending their ecological and evolutionary adaptations. By using a new tracking method to extract locomotory data from free-living marine fish, we have opened up new possibilities for studying the biology, behavior, and ecology of many other species in their natural conditions. Furthermore, this tracking method can help researchers identify how different environmental factors, such as temperature and light, affect the circadian rhythm of activity–rest in marine animals. This information can be used to predict the response of marine animals to changes in their environment and, ultimately, help guide conservation efforts. We found that the activity–rest circadian rhythm of Xyrichtys novacula was well synchronized by the environmental cycle of light–darkness regardless of gender and the period studied. However, the reproductive season caused direct effects on the rhythm, resulting in increased fragmentation and decreased synchronization in both genders.

**Abstract:**

Recent technological advances in marine biotelemetry have demonstrated that marine fish species perform activity–rest rhythms that have relevant ecological and evolutionary consequences. The main objective of the present report is to study the circadian rhythm of activity–rest of the pearly razorfish, *Xyrichtys novacula* in its own habitat, before and during the reproduction season using a novel biotelemetry system. This fish species is a small-bodied marine species that inhabits most shallow soft habitats of temperate areas and has a high interest for commercial and recreational fisheries. The activity of free-living fish was monitored by means of high-resolution acoustic tracking of the motor activity of the fish in one-minute intervals. The obtained data allowed the definition of the circadian rhythm of activity–rest in terms of classical non-parametric values: interdaily stability (IS), intradaily variability (IV), relative amplitude (RA), average activity during the most-active period of consecutive 10 h (M10), and average activity during the least-active period of consecutive 5 h (L5). We observed a well-marked rhythm, with little fragmentation and good synchrony with the environmental cycle of light–darkness, regardless of sex and the period studied. However, the rhythm was found to be slightly more desynchronized and fragmented during reproduction because of variations in the photoperiod. In addition, we found that the activity of the males was much higher than that of the females (*p* < 0.001), probably due to the peculiar behavior of the males in defending the harems they lead. Finally, the time at which activity began in males was slightly earlier than it was in females (*p* < 0.001), presumably due to the same fact, as differences in activity or for the individual heterogeneity of this species in the time of awakening are considered to be an independent axis of the fish’s personality. Our work is novel, as it is one of the first studies of activity–rest rhythm using classical circadian-related descriptors in free-living marine fish using locomotory data facilitated by novel technological approaches.

## 1. Introduction

Natural selection has allowed many organisms to develop structures related to the circadian system, thus allowing them to establish and synchronize physiological responses in periods approximated to 24 h, thereby increasing their survival [1]. The circadian system possesses three main elements: the environmental synchronizers, the circadian clock or oscillator, and the output pathways [2].

In all animals, the circadian system is composed of a multitude of oscillators located in the brain and tissues. In fish, the hierarchy in oscillators is not as evident as it is in mammals, where the suprachiasmatic nucleus functions as a master clock that is capable of synchronizing and regulating the rest of the clocks.

Teleosts do not have a single master clock in the same way that mammals do. Instead, their circadian rhythms are regulated by a distributed system of clocks that are found in various organs and tissues throughout the body, including the eyes, the brain, and the pineal gland. The peripheral clocks appear to be able to entrain to other zeitgebers, such as feeding time [3]; this indicates independent sources of synchronization [4].

In many teleosts, the pineal gland is considered to be the primary pacemaker or oscillator of the circadian system, producing the hormone melatonin in a rhythmic manner in response to changes in light and dark cycles [5]. The pineal gland of teleosts, at least in zebrafish, has been shown to receive input from the eyes and other brain regions [5,6], which help to synchronize its rhythm with external cues.

In addition to the pineal gland, other regions of the brain have also been implicated in regulating the circadian rhythm of teleosts. For example, the hypothalamus, which is involved in the regulation of many physiological processes, including temperature regulation, has been shown to contain circadian clock genes in some teleost species [7].

Summarizing, the circadian rhythm of teleosts is regulated by a complex and distributed system of clocks, with the pineal gland playing a prominent role in many species.

The role of environmental factors, such as photoperiod and temperature, on different aspects of teleost physiology, including the annual reproductive cycle, is well known [8,9]. The hypothalamohypophyseal system is regarded as a possible mediator of the environmental information received via the pineal organ or the lateral eyes, transducing it into appropriate neuroendocrine signals to produce metabolic, physiological, and behavioral alterations [10].

The hypothalamus plays a critical role in regulating temperature and other physiological processes in teleosts [7], helping them to maintain homeostasis and adapt to changes in their environment. One of the key responses to changes in temperature is the regulation of the metabolic rate. Teleosts are ectothermic, meaning that their body temperature is largely determined by the temperature of their environment. As a result, changes in the temperature of the water or the air can have a significant impact on their metabolic rate, which in turn affects their overall physiology and behavior.

Temperature cycles, also known as thermocycles, are important synchronizers of circadian rhythms in many organisms. Like light cycles, temperature cycles can influence the expression of clock genes and the timing of circadian activity rhythms [11]. In constant conditions, zebrafish, for example, showed significant free-running rhythms, which indicate that circadian locomotor activity was entrained to the thermocycles and was not a result of masking [12]. Zebrafish synchronized mostly to the light cycle, although they displayed relative coordination, as their locomotor activity increased when light and thermophase coincided. The light phase coincides with the thermophase (the phase of higher temperature) and the dark phase coincides with the chryophase (the phase of lower temperature); thus, transitions from cold to warm temperature are roughly associated with dawn, and transitions from warm to cold temperature are roughly associated with dusk [13].

Although light is a stronger synchronizer than temperature, thermocycles alone can entrain circadian rhythms and interfere in their light synchronization, suggesting the existence of both light- and temperature-entrainable oscillators that are weakly coupled [11]. While temperature can entrain fish locomotor activity rhythms, it is generally considered to be a weaker zeitgeber than light. This is because temperature changes are often slower and less predictable than changes in light, and because fish may have different thermal preferences, depending on their species and habitat. In sea bass, the plasma melatonin rhythms are primarily driven by the photoperiod length, which determines the duration of the nocturnal rise in melatonin levels, and by water temperature, which determines the amplitude of the rhythm [14]. However, under natural conditions, the pineal organ in sea bass has been shown to have the ability to integrate seasonal information, such as changes in temperature, the photoperiod, and other environmental cues. This integration of information allows the pineal gland to generate a specific melatonin profile for each season, which can provide precise calendar information to the fish [14].

The pearl razorfish, Xyrichtys novacula, is a small-bodied fish that inhabits the shallow waters with sandy bottoms and seagrass of tropical and temperate seas [15] This species, the only representative of the genus Xyrichtys in the Mediterranean Sea, is of high interest for commercial and recreational fishing. It can be found at depths between 0 and 50 m and is a diurnal species that buries itself in the sand to rest at night. Its diet consists of small benthic invertebrates, such as molluscs, crustaceans, echinoderms, and polychaetes [16] The pearl razorfish is a protogynous monandrous hermaphrodite with sexual dimorphism, and males are highly territorial, defending their harems from other males [17]. The reproductive period of this species takes place during the summer, and only the dominant males participate in spawning [18].

Considering these preliminary observations, the main objective of the present report is to study the circadian rhythm of activity–rest in the species *Xyrichtys novacula*, before and during the reproduction season, in natural conditions. The data used in this study are based on locomotory data (distance travelled per minute) that were extracted using a novel method of positioning of free-living marine fish, allowing the application of classical approaches to study the circadian rhythms applied to laboratory studies. As a departure hypothesis, we assumed that the reproductive season of *Xyrichtys novacula* significantly alters the expression of its circadian rhythm of activity and rest. In addition, it was expected that its biological rhythms would show gender-related variations in the rhythms of rest/activity. In specific terms, our objectives were (i) to define the activity–rest circadian rhythm of *Xyrichtys novacula*, considering the non-parametric variables currently used to characterize the rhythm, (ii) to determine the influence of the reproductive season in the manifestation of the circadian rhythms, and (iii) to determine the influence of gender in the manifestation of the circadian rhythms.

## 2. Materials and Methods

### 2.1. Experimental Animals

We used data obtained from a previous acoustic monitoring experiment, in which 58 specimens of the species *Xyrichtys novacula* (28 females with an average total length of 15.31 ± 0.81 cm and 30 males with an average total length of 19.64 ± 1.09 cm—total length measured using and icterometer) were captured and tracked in the waters of the marine protected area of Palma Bay (Mallorca Islands, NW Mediterranean, [19]. The pearly razorfish, *Xyricthys novacula*, is a small-bodied, territorial wrasse that inhabits shallow sandy bottoms of temperate waters of the Atlantic Ocean, the Mediterranean Sea, and the Caribbean Sea [19]. The species shows sexual dimorphism, which allows individuals to be visually sexed. They live in harems formed by a male and several females [20]. Natural populations of this species are formed by individuals displaying a rich repertoire of behavioral types [21]. This species displays burrowing behavior; the fish emerge from the sand in the morning, forage during the daytime, and bury themselves in the sand during the nighttime to rest and escape from nocturnal predators [21]. This behavior makes this species a perfect model system to characterize activity–rest rhythms, as their activity patterns are relatively easy to monitor through acoustic telemetry. This study comprised four groups: (1) males in a pre-reproductive state, (2) males during a reproductive state, (3) females in a pre-reproductive state, and (4) females during a reproductive state.

### 2.2. Placement and Conditions of Study

The measurement of the activity of the fish was carried out during the spring and summer months of 2019. We analyzed two periods of ~20 days. The first measurement was made during the pre-reproductive season, from 13 May to 2 June; a second measurement was made during the reproductive season, from 1 to 21 July [17,20].

The water temperature was assessed every 10 min by a recorder located in the center of the study area at a depth of about 14 m, resulting in an average temperature of 18.30 ± 0.64 °C and 24.52 ± 1.56 °C during the first and second seasons, respectively. The hours of daily light were also measured during the first and second seasons, resulting in ~14 h 30 min and ~14 h 45 min of light, respectively. Light and temperature were measured with a Hobo Pendant^®^ (Pendant Temperature/Light 64 K Data Logger) placed within the receiver’s array.

The experimental area was placed in the marine reserve of the Bahía de Palma; more specifically, a sandy area near the coast in the closed area of the reserve was selected (see details in [19]). This area was chosen because it met a series of characteristics that were suitable for this study, such as being relatively small (600 m by 270 m, (12.5 ha) and a depth between 11–19 m). The area was delimited by a *Posidonia oceanica* meadow that functions as a natural barrier, which limited the mobility of the individuals. In addition, the area is inhabited by a large population of *Xyrichtys novacula* [21].

### 2.3. Monitoring the Locomotor Activity

Monitoring was carried out according to the implementation described in Aspillaga et al. [19]. Briefly, fish were tagged with micro-transmitters that emitted an acoustic signal (at 416.7 kHz) that allowed for the identification of the individual (ID), and, through an array of listening stations (receivers) and a triangulation and post-processing algorithms, allowed for positioning (in latitude and longitude) every minute (on average) with high accuracy and precision. Thus, after the capture of the individuals using standardized fishing gear, and following all the pertinent animal care protocols, and after having obtained the necessary permits from the Ethics Committee of the Universitat de les Illes Balears, each specimen was submerged in a 0.1 g/L solution of the anesthetic tricaine methanesulfonate. Microtransmitters were implanted into the peritoneal cavity through a small ventral incision, which was then closed with non-resorbable sutures. Prior to release, the specimens were placed in a tank with clean seawater until they recovered their normal behavior. The activity of the fish was measured as the distance in meters that the fish travelled per minute using the geographical positions (latitude and longitude) that the tracking systems produced (see Section 2.3). By selecting this type of small size transmitter, almost all size ranges in the fish could be studied, unlike previous similar systems. The experiment considered a total of 70 receivers placed at a distance of 50 m, forming an equilateral triangular pattern in our study area [19]. An algorithm was developed to triangulate the exact position of the fish carrying the transmitter by capturing at least 3 receivers of the acoustic signal emitted by the micro-transmitter. Thus, with this technology, multiple fish could be tracked at the same time, since it was capable of picking up signals emitted between them with a time of approximately 1 millisecond at a distance of up to 1 km in the open sea. The obtained raw trajectories were post-processed by applying a trajectory filter to remove the most obvious outliers and a continuous-times-correlated random-walk-movement model [19] to generate with regular timestamps [21]. The monitoring system was extensively calibrated and validated before recording the data for our study site [19]. From the positions generated, we created a temporal time series, for each individual, of locomotor activity (distance travelled per unit of time) per each individual (Appendix A).

### 2.4. Analysis of Data

#### 2.4.1. Visual Analysis

Actograms of the individual locomotory data (m travelled per min) were plotted for each individual to visualize activity–rest patterns. *X. novacula* suffered some days of anormal behavior by remaining buried in the sand as a consequence of the surgery [18], and these days were removed from the dataset. The actogram is a graphic representation that records biological variables such as circadian rhythms. The actograms of individual tike series of activity were elaborated using the OriginPro program. The vertical axis provided information about the period (in hours) and about the rhythm, while the horizontal axis showed the activity within each cycle [22].

Therefore, the actogram provided an orderly visualization of the data and allowed the study of the entrainment and the phase shift of the light–dark cycle rhythms.

#### 2.4.2. Non-Parametric Analysis of the Variables Defining the Circadian Rhythms

The rhythm of rest–activity was studied with a non-parametic approach [23] using the “nparACT” package of the R software (RStudio 2021.09.01) [24]. The non-parametric variables obtained were as follows:(a)Interdaily stability (IS), which ranged from 0 to 1. The IS quantified the rhythm stability on different days and was more dependent on external factors or zeitgebers. The higher the value, the better the synchronization.(b)Intradaily variability (IV), which was a measure of the fragmentation of the rhythm ranging between 0 (no variability) and 2 (high fragmentation).(c)Relative amplitude (RA), which ranged between 0 and 1. It depended on internal and external influences together [25] and was calculated from the average activity during the most active period of 10 consecutive hours (M10) and the average activity during the least active period of 5 consecutive hours (L5): (1)(d)The software package offered the values of both L5 and M10, as well as the average time in which both periods started, and(e)The circadian function index (CFI), which showed the robustness of the rhythm. CFI was calculated by considering together the IV, IS, and RA. It ranged between 0 (low robustness, without circadian rhythm) and 1 (high robustness) [26].
RA = (M10 − L5)/(M10 + L5)(1)

#### 2.4.3. Activity Rhythm Analysis

We studied the activity data during the pre-reproductive and reproductive season. We used the OriginPro program to define the activity of the fish to obtain the following indices that may define the rhythms:− Maximum activity (m), which was the maximal length (m) of intervals of movement recorded in one-minute intervals, and− Average activity per cycle (m/day), which was the daily average number of meters traveled by the fish in a complete cycle or day.

We used the program Origin-Pro to analyze, in each of the four described groups, the schedules of the average activity–rest rhythm. In addition, we used the Poincare diagram of the Origin-Pro program to graph the patterns of movement in each of the four described groups. The Poincaré diagram is a type of scatter plot used in the analysis of complex systems, often in the context of time series data. Origin-Pro was also used to create a two-dimensional 95% confidence ellipse, a statistical tool to analyze the activity patterns of the animals during the pre-reproductive and reproductive seasons.

#### 2.4.4. Temperature Rhythm Analysis

OriginPro software (Version 2023) was used to analyze water temperature changes during two different recording periods (before and during reproduction). We calculated the trend line and the Pearson correlation coefficient between the water temperature and the successive days of recording. We also analyzed the interaction between the temperature of the sea water and the activity of this species, by calculating the correlation coefficient of Pearson.

#### 2.4.5. Statistical Analysis

The activities of the 58 individuals were contrasted using program R to determine the significance of the differences between the four groups of individuals:(1)Males in a pre-reproductive period were compared with males in a reproductive period. Since the same individuals were studied in different times, the statistics were performed for paired samples.(2)Females in a pre-reproductive period were compared with females in a reproductive period. As previously described, the statistics were performed for paired samples.(3)Males and females studied in a pre-reproductive period. Since these two groups contained individuals paired in a form different from that of the previous groups, the statistics were performed for unpaired samples.(4)Males and females were compared in a reproductive and a pre reproductive state, respectively. As in the previous paragraph, the statistical difference was performed for unpaired samples.

The statistical test that was used depended on whether the values obtained met the parameters necessary to perform that test. For both the paired and unpaired groups, all data were analyzed for normality (based on the Shapiro–Wilk statistical test); when the samples were normal, we used the Student’s t-test. For non-normal samples, we used the Wilcoxon test in the paired groups and Mann–Whitney tests in the unpaired groups. We also analyzed the homogeneity of variances for both groups using the Bartlett test. If both the normality and the homogeneity of variances were met, the Student’s t-test was performed for unpaired samples. When the variances were normal but not homogeneous, we used the Welch correction t-test. The minimum significance level for all statistical tests was 0.05.

## 3. Results

### 3.1. Actograms of the Rest–Activity Rhythm of Xyrichtys Novacula in Pre-Reproduction and Reproduction Periods

Many fish species exhibit a daily pattern of activity and rest that is synchronized with the light–dark cycle. This rhythm is controlled by an internal biological clock that is reset each day by environmental cues such as light and temperature.

Figure 1 shows Actograms of the average activity-–rest cycle for each of the four *Xyrichtys novacula* groups. The activity rhythm of the fish of both sexes is driven by the 24-h ambient light–dark cycle during the pre-reproduction and reproduction periods.

### 3.2. Non-Parametric Analysis of the Variables Defining the Circadian Rhythms

In addition, the non-parametric indices defining the circadian activity–rest rhythm demonstrated the existence of a significant light/dark rhythm. The mean value of the IS before and during the reproduction period, in males and females, was closer to 1 than to 0. This indicated that the circadian rhythm was quite synchronized in both sexes and in both periods. The IS values in males showed a statistically significant decrease (*** *p* < 0.01) during the reproduction season, from 0.867 ± 0.013 to 0.764 ± 0.015, which indicated a higher desynchronization of the rhythm during the reproduction period (Figure 2). In females, no statistical differences were observed between the IS values recorded before and during reproduction (Figure 2), which were 0.707 ± 0.026 and 0.710 ± 0.020, respectively. When comparing the sexes during the pre-reproduction period, females presented an average IS value (0.707 ± 0.026) that was statistically lower (*p* < 0.001) than that of males (0.867 ± 0.013). This meant that the circadian rhythms of females were slightly impaired during this period. However, the difference between the sexes disappeared during reproduction, being slightly higher in males (0.764 ± 0.015), but without differences with females (0.710 ± 0.020) (Figure 2)

We observed that the average IV in males and females before and during the reproductive period was closer to 0 than to 2. This indicated that the circadian rhythm was not fragmented in both sexes and in periods. However, we observed a significant increase (*p* < 0.001) in the IV of males during the reproduction season (Figure 3), from 0.322 ± 0.009 to 0.391 ± 0.017, which indicated a greater fragmentation of the rhythm during reproduction. In females, we observed an increase in IV during the reproductive period, but the differences were not statistically significant (average IV: 0.452 ± 0.036 in pre-reproduction and 0.472 ± 0.025 during reproduction) (Figure 3).

When comparing the sexes, during the pre-reproduction period, the previous figures showed that the IV of females presented an average IV value that was statistically higher than that of males (*p* < 0.001). The same occurred during the reproductive period, in which females showed a higher average IV than males (*p* < 0.05) (Figure 3).

RA, which depends on the values of M10 and L5, as explained above, showed the same value (RA = 1) in all the analyzed fish. This was due to the fact that, in the wild state, individuals of this species bury themselves throughout the night, i.e., the value of L5 is null. Therefore, considering that RA = (M10 − L5)/(M10 + L5) and that L5 = 0, the value of AR is 1.

As explained before, CFI integrates normalized values between 0 and 1 for IS, IV and RA, inverting the IV values (CFI = (IS + RA + ((2−IV)/2))/3). Null circadian rhythmicity is considered when CFI is equal to 0, and maximally robust when its value is equal to 1. The mean value of CFI before and during the reproduction period, in males and females, was closer to 1 than to 0, thus indicating that the circadian rhythm was quite robust in both sexes and in both periods (Figure 4).

Differences were observed, between genders and in the periods analyzed, in the M10 values and in the start time of these 10 h of maximum activity. The average value of M10 in males did not show differences when comparing the periods before and during reproduction (Figure 5); the average value went from 5.240 ± 0.529 m/min to 5.233 ± 0.401 m/min.

In the case of females, differences were observed between both periods but they were not statistically significant; the average value before and during reproduction was 1.217 ± 0.118 m/min and 0.913 ± 0.086 m/min, respectively (Figure 5). When comparing the sexes during the pre-reproduction period, the females presented an average value of M10, which was statistically lower than that of the males (Figure 5). The same occurred during the reproduction period, presenting a higher mean value of M10 in males than in females.

The start time M10 gives an approximate idea of the time when the individuals start their daily activities. In male individuals, the start time M10 was moved from being approximately 7:21 during the pre-reproduction season to around 8:14 during the reproduction season; these were statistically different values (*p* < 0.001). In the case of females, no difference was found with an M10 starting time at approximately 9:00 in the morning (Figure 6). When comparing the sexes, the start time of M10 was delayed in females, both in the pre-reproductive and reproductive periods (Figure 6). When comparing the sexes, the onset time of M10 was delayed in females in both the pre-reproductive and reproductive periods (Figure 6). The difference reached significance during the pre-reproduction periods (*p* < 0.001) and reproductive periods (*p* < 0.05).

Regarding the start time of L5 during the pre-reproduction and reproduction periods, we found statistically significant differences between both males (*p* < 0.001) and females (*p* < 0.001). However, no significant differences of L5 were observed between males and females in the same period (Figure 7).

### 3.3. Maximum Activity and the Mean Activity Per Cycle

Regarding maximum activity, statistically significant differences were observed in both males and in females. In males, the maximal activity was higher (*p* < 0.05) during the reproductive period (3.047 ± 0.242 m/min) than during the pre-reproductive period (2.373± 0.145 m/min). In females, the maximal activity was higher (*p* < 0.001) during the pre-reproduction period (1.605 ± 0.184 m/min) than during the reproductive period (0.804 ± 0.093 m/min) (Figure 8).

When comparing the sexes, the average maximal activity was statistically higher in males, both in the pre-reproductive (*p* < 0.05) and reproductive periods (*p* < 0.001), than it was in females (Figure 8).

Regarding the average value of activity per cycle (m), no statistically significant differences were observed between males during the pre-reproductive (392.125 ± 36.833 m) and reproductive periods (407.662 ± 45.642 m) (Figure 9). However, in females, the mean activity per cycle was statically higher (*p* < 0.001) during pre-reproductive periods (84.505 ± 10.719 m) than in reproductive periods (54.136 ± 5.319 m). Finally, when comparing the sexes, the average activity per cycle was also statistically higher in males, both in the pre-reproductive (*p* < 0.001) and reproductive periods (*p* < 0.001) (Figure 9). When comparing between sexes, the average activity per cycle was also statistically higher in males, both in the pre-reproductive and reproductive periods (Figure 9).

### 3.4. Distribution of Activity in Pre-Reproductive Periods and Reproduction

Finally, the average values of the displacement in each of the measurements are graphically represented by means of the Poincare diagram. The two-dimensional 95% confidence ellipse shows that males travel greater distances in both periods, as the ellipses for males are larger than those for females. This indicates that males are more active and cover more ground than females during pre-reproductive periods and reproduction periods (Figure 10).

### 3.5. Water Temperature during the PP and RP

The average water temperatures recorded at each hour of the day followed a cyclical pattern during PP and RP (Figure 11). The mean water temperatures were highest during the day and lowest at night, indicating a circadian pattern of temperature variation. The average water temperatures during RP were higher than those during PP, due to changes in environmental conditions or weather patterns (change from spring to summer).

The water temperature was compared in successive days during two different periods (Figure 12): PP (from 13 May to 2 June) and RP (from 1 July to 21 July). The correlation coefficient of 0.61 during PP indicates a moderately strong positive correlation between water temperature and successive days of recording during that time period. On the other hand, the correlation coefficient of 0.37 during RP indicates a weaker positive correlation during that time period.

Figure 13 shows the Pearson correlation coefficient between water temperature and fish activity during two different time periods, PP and RP, for male and female fish separately. During PP, the correlation coefficient between water temperature and fish activity is 0.26 in males and 0.31 in females, indicating a positive but weak correlation in both genders. During RP, the correlation coefficient is 0.038 in females and 0.084 in males, indicating an even weaker positive correlation in both genders.

## 4. Discussion

Here, we report the activity rhythm in a free-living marine fish species by means of acoustic tracking. A clear circadian rhythm in the motor activity of pearly razorfish was found regardless of the sex and the period (before or during reproductive season). The activity in its natural habitat was determined by a marked circadian rhythm of activity–rest that was well synchronized by the environmental cycle of light–darkness.

We found IV values close to 0 and IS and CFI values close to 1 in all the analyzed groups. In addition, the RA was unchanged in all fish. This demonstrated that circadian rhythm is robust and well synchronized with the environment in all cases and shows a reduced fragmentation, confirming the presence of well differentiated rest/activity periods. Furthermore, the activity decreased to zero during the resting phase (L5 = 0 m/min in all fish), coinciding with the characteristic behavior of the species, which burrows under the sandy bottom during rest. This diel behavior was previously demonstrated in this species using the acoustics detection patterns of the individuals [18]. However, thanks to the generation of accurate locomotory data (displacement per min), the present report shows that the activity of the pearly razorfish is precisely dependent on the environmental cycle of light–darkness.

Regarding the differences in the rhythm of males and females that were found before and during the reproduction period, the rhythm was better synchronized and less fragmented in males. Furthermore, we found that the reproductive activity caused direct effects on the rhythm, provoking light improvements in synchronization and increased fragmentations. Indeed, during the pre-reproductive period, the synchronization was statistically greater in males but, when approaching the reproductive season, the rhythm was slightly desynchronized, matching with the desynchronization of females, which was maintained with no changes in the two periods. Similar results were observed in the fragmentation of the rhythm that was greater in both periods. Nevertheless, the fragmentation increased in both sexes during reproduction. Several studies observed similar differences between sexes maintained under stable laboratory conditions (12/12 L/D), for example, in the activity patterns of males and females of Nile Tilapia (*Oreochromis niloticus*) [27]. In addition, it is known that the activity–rest patterns of fish show great plasticity [28,29].

The differences in the circadian rhythm of activity–rest between periods may be due to environmental factors. In general, the reproduction of most fish is seasonal and periodic. Reproduction depends on an endogenous rhythm following a seasonal clock, although it is also dependent on the photoperiod. In this case, the resulting rhythm is circannual, associated to the cyclic pattern of melatonin, guaranteeing reproduction during the most suitable season for the subsequent development of new individuals. Several studies have shown the existence of different patterns in the nocturnal secretion of melatonin in fish, even in the same individual, depending on the reproductive season [30]. Therefore, it is considered that the variations in the circadian rhythm of activity–rest observed could be due to the progressive increase in daylight hours during the two periods (on average, approximately 14 h and 30 min during pre-reproductive phases and 14 h 45 min during reproductive ones), or to a possible change in the nocturnal pattern of melatonin secretion when entering the reproduction season, which would influence the rhythm of activity–rest, as melatonin is one of the factors that regulate both rhythms.

The existence of both light and temperature entrainable oscillators that are weakly coupled refers to the presence of two separate biological rhythms, one influenced by light and the other influenced by temperature, which interact with each other to a limited extent [11]. In razorfish, the Pearson correlation coefficient between water temperature and activity shows a positive but weak correlation in both sexes during PP and an even weaker correlation during RP. For many aquatic poikilothermic animals, such as fish, light is a crucial factor that can influence their behavior, reproduction, and development [31]. Light can affect the timing of daily activities, such as feeding and spawning, as well as the migration patterns of certain species. Additionally, light is necessary for the development of pigments that provide camouflage and other forms of visual communication [31]. Overall, while temperature and light are important factors for both terrestrial and aquatic poikilothermic animals, the specific importance of each factor may vary, depending on the species and its environment.

Regarding the activity that is understood as the displacement of the fish, we observed that the start time of L5 was statistically the same in males and females, and only varied with the period studied. This time, without being the start time of the rest phase, could be considered as something similar. The differences observed between the periods studied were probably due to the increase in the photoperiod, that is, the delay in sunset that occurred between the two measurements. During the measurement in the pre-reproduction phase, sunset occurred approximately, on average, at 21:05, while in reproduction it occurred, on average, at 21:20. This 15 min delay also occurred in the start time of L5. In both sexes, the L5 start time was 21:15 and 21:30 in pre-reproduction and reproduction, respectively.

Accordingly, it could be said that the beginning of rest is directly related to sunset. In the case of the start time of M10, which can be related to the start time of the activity, the observed differences no longer fit the sunrise time. In this case, for both periods, dawn occurs at practically the same time, being on average at 6:30, so the delay in females compared to males during pre-reproduction and the delay in males at entering reproduction probably come from causes intrinsic to the animal.

In all activity of the fish values analyzed, as represented in the actograms and the Poincare distribution diagrams, the displacement of the males was much higher than that of the females. In addition, observing the calculated activity values, when passing to the reproduction period, the activity of the females slightly decreased, while that of the males increased slightly.

It is common for water temperatures to follow a cyclical pattern during the day, with higher temperatures during daylight hours and lower temperatures at night. This is because sunlight warms the water, while at night, the water cools down. The average water temperatures during RP were higher than the average water temperatures during PP, which may be due to changes in environmental conditions or weather patterns. This is likely because during summer, the weather is generally warmer and there is more sunlight, which can increase water temperatures. During PP, the correlation coefficients between water temperature and fish activity were 0.26 in males and 0.31 in females, indicating a positive but weak correlation in both genders. This means that as water temperature increased, fish activity tended to increase slightly, but the relationship between the two variables was not particularly strong. During RP, the correlation coefficients were even weaker, with a coefficient of 0.038 in females and 0.084 in males. This suggests that there was little to no relationship between water temperature and fish activity during the summer months, at least as measured by the correlation coefficient. It is important to note that correlation does not necessarily imply causation. While the correlation coefficients suggest a weak positive relationship between water temperature and fish activity during PP and a weaker relationship during RP, other factors could be affecting fish behavior and activity levels as well, such as the availability of food, predation risk, and breeding behaviors.

The mechanisms involved in behavioral thermoregulation in fish are still not fully understood [32], but it is thought that the temperature detection system plays a fundamental role in the behavioral response. Fish are ectothermic; as a result, changes in water temperature can have significant impacts on fish physiology and behavior. To avoid physiological damage caused by acute increases or decreases in water temperature, fish may engage in behavioral thermoregulation, which involves actively seeking out areas of water with temperatures that are more suitable for their physiological needs [33].

As in the case of tilefish [34], the formation of harems in pearly razorfish, with a dominant male and a group of females, is another important aspect of their social behavior [20,35]. The dominant male exercises great territorialism against other males, which is necessary to protect his harem and ensure reproductive success.

The behavior and social structure of the pearly razorfish suggest that aggression is a critical component of their social interactions and territorial defense. The facts that both males and females exhibit aggressive behavior toward each other and maintain separate territories suggest that aggression is not limited to a specific gender and is, instead, an inherent aspect of their species’ behavior [36]. This is a common pattern in many animal species, where individuals may exhibit different levels of aggression depending on their sex, social status, or reproductive state.

Male razorfish are more active than females, which could be attributed to structural differences rather than to behavior alone. For example, males may have larger bodies or more developed muscles, which enable them to be more active and aggressive in their territorial defense. Overall, the aggressive behavior of the pearly razorfish, regardless of gender, highlights the importance of social dynamics and territorial boundaries in this species. By maintaining separate territories and engaging in aggressive behavior, these fish can secure their resources and ensure their survival in their natural habitat.

In addition to these personality traits, this species tends to be quite sedentary, with individuals typically remaining within a small area of less than 0.5 km^2^ [21]. This suggests that the species may have a strong preference for a particular habitat or environment, or that they may be territorial and defend a specific area against other individuals. Overall, these personality and behavioral traits may have important implications for the ecology and social dynamics of this species and may play a role in shaping their interactions with other individuals and species within their environment.

In summary, the social behavior of pearly razorfish, including their aggressive behavior, territorialism, and formation of harems, is related to the differences between the sexes observed. These behaviors are essential for their survival and reproductive success in their natural habitat.

Finally, the differences observed in the start time of M10, which is early in the males, would also indicate the need for the male to defend his harem by starting this defense activity just before the females become active. The slight differences observed between the sexes in the circadian rhythm of activity–rest are possibly due to these behavioral differences.

## 5. Conclusions

Our work shows a marked activity–rest circadian rhythm of the fish *Xyrichtys novacula* that is little fragmented and well synchronized with the environmental light–dark cycle, regardless of sex and the season studied, where activity occurs during the day and rest occurs at night. During the reproduction season, the circadian rhythm of activity–rest of these fish is slightly more desynchronized and fragmented, compared to the pre-reproductive period, which is probably caused by the increase in the photoperiod and its consequent slight modification of the nocturnal pattern of melatonin secretion. This study also shows how the circadian rhythm of activity–rest in males is slightly more synchronized and less fragmented than it is in females, possibly due to behavioral differences between the sexes. The activity, understood as the displacement, of the males, is representatively higher than it is in the females, regardless of the period studied. This is due to the aggressive and territorial behavior of the males, and their need to protect and watch over their harem in front of other males. The start time of rest is determined by the environmental cycle of light–darkness, specifically with dusk. However, the time at which activity begins in males is slightly earlier than it is in females, presumably due to the same fact, as differences in activity or for the individual heterogeneity of this species in the time of awakening are considered to be an independent axis of the fish’s personality. This work is one of the first studies of the activity–rest circadian rhythm in free-living marine fish and shows that high-resolution acoustic tracking is a proper methodology to extract activity data to further study activity–rest circadian rhythm in marine fish.

## Figures and Tables

**Figure 1 biology-12-00810-f001:**
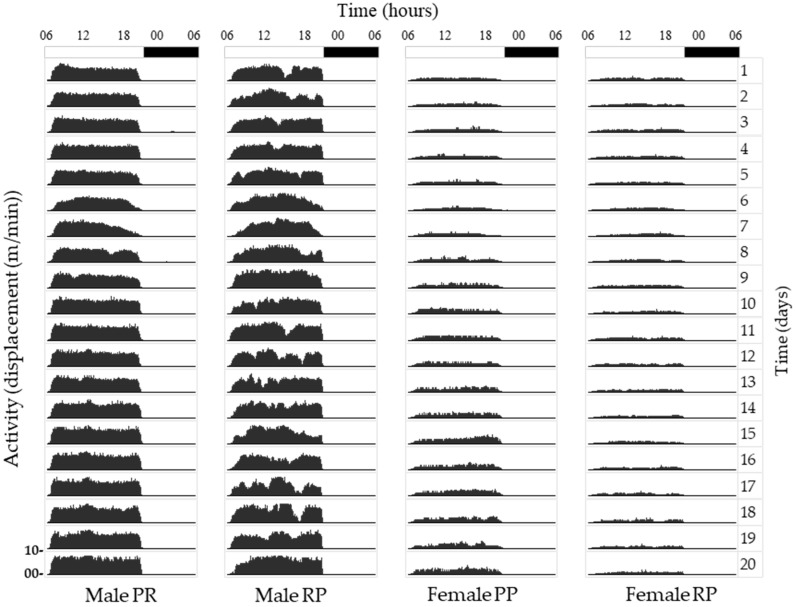
Actograms of the average activity–rest cycle for each ”protoptypical” individual of the four groups (a male and female during the pre-reproductive (PP) and reproductive period (RP). The first bar represents the hours of darkness (in black). Note that regardless of the period, males always showed higher activity than females.

**Figure 2 biology-12-00810-f002:**
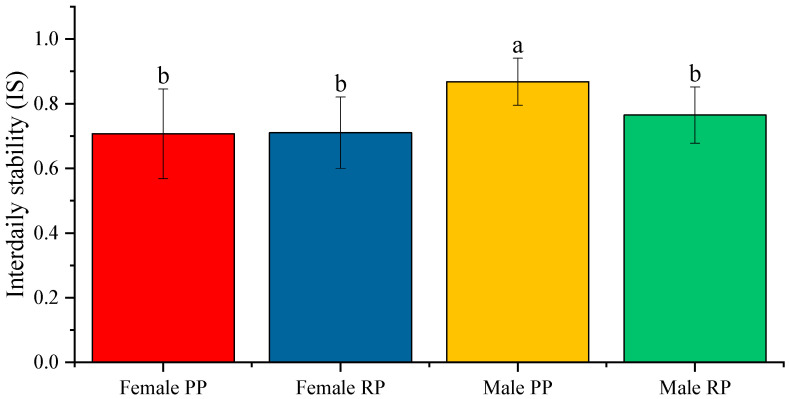
Average value of the interdaily stability (IS) in male and female fish before the reproductive period (PP) and during the reproductive period (RP). The data represented are the mean of each group ± SD; *n* = 28 female fish and 30 male fish. Different letters indicate statistically significant differences. The intervals above each bar represent the standard deviation.

**Figure 3 biology-12-00810-f003:**
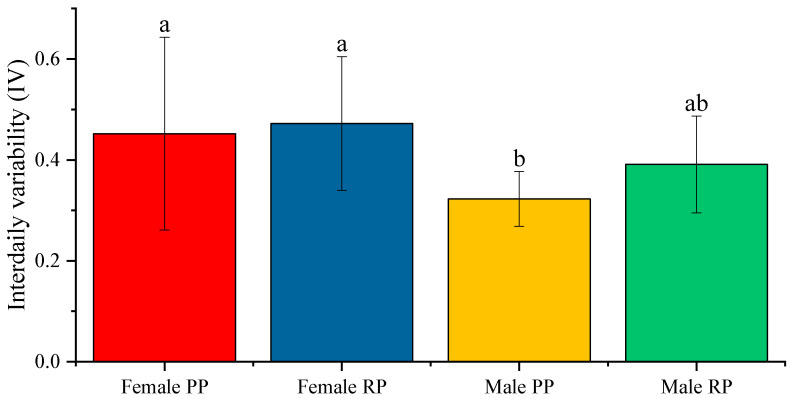
Average of the intradaily variability (IV) of the activity–rest circadian rhythm in male and female fish before and during the reproductive period. The data represented are the mean of each group ± SD, *n* = 30 male fish and 28 female fish. Different letters indicate statistically significant differences. The intervals above each bar represent the standard deviation.

**Figure 4 biology-12-00810-f004:**
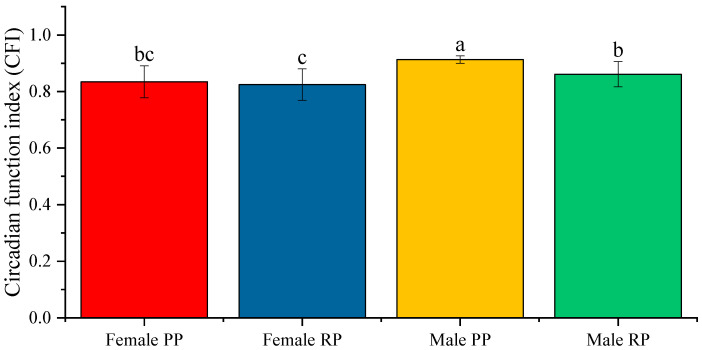
Average value of the CFI in male and female fish before and during the reproductive period. The data represented are the mean of each group ± SD, *n* = 30 male fish and 28 female fish. Different letters indicate statistically significant differences. The intervals above each bar represent the standard deviation.

**Figure 5 biology-12-00810-f005:**
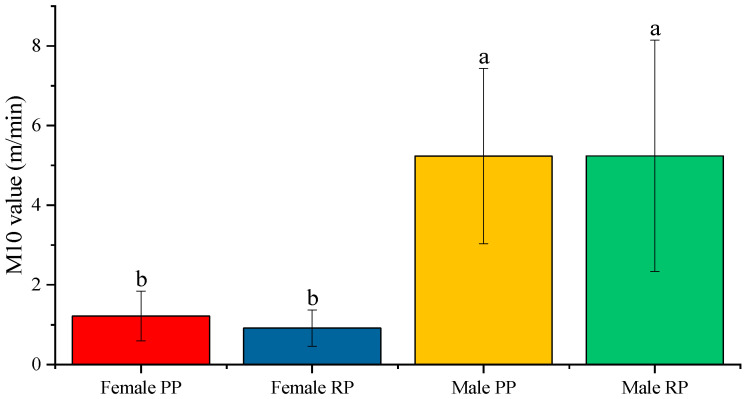
Average value of the mean activity in the 10 h of maximum activity (M10) of the activity–rest circadian rhythm in male and female fish before and during the reproductive period. The data represented are the mean of each group ± SD, *n* = 30 male fish and 28 female fish. Different letters indicate statistically significant differences. The intervals above each bar represent the standard deviation.

**Figure 6 biology-12-00810-f006:**
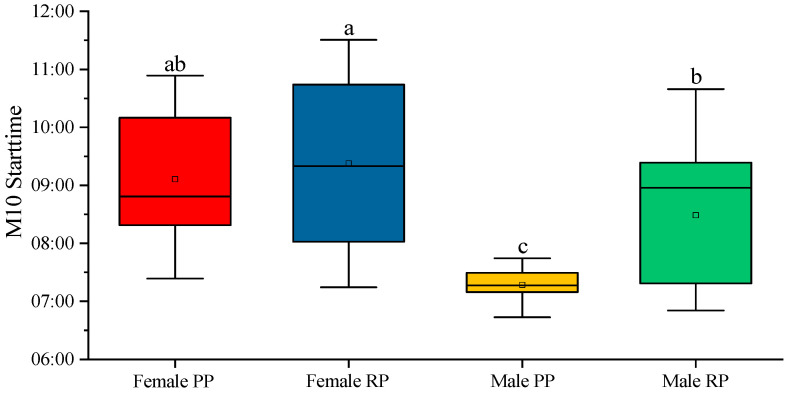
M10 start time of the activity–rest circadian rhythm in male and female fish before and during the reproduction period. The data represented are the mean of each group ± SD, *n* = 30 male fish and 28 female fish. Different letters indicate statistically significant differences. The intervals above each bar represent the standard deviation.

**Figure 7 biology-12-00810-f007:**
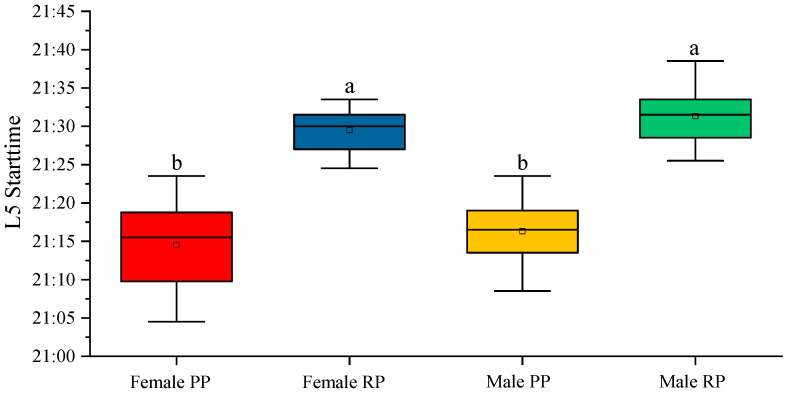
L5 start time of the activity–rest circadian rhythm in male and female fish, before and during the reproduction period. The data represented are the mean of each group ± SD, *n* = 30 male fish and 28 female fish. Different letters indicate statistically significant differences. The intervals above each bar represent the standard deviation.

**Figure 8 biology-12-00810-f008:**
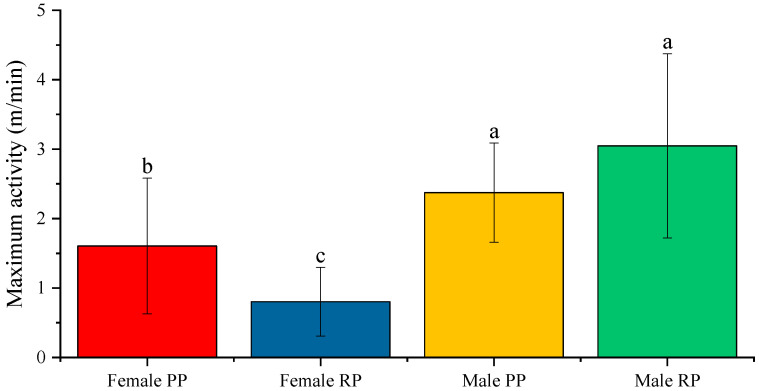
Average value of maximum activity (m/min) in male and female fish before and during the reproductive period. Different letters indicate statistically significant differences. The intervals above each bar represent the standard deviation.

**Figure 9 biology-12-00810-f009:**
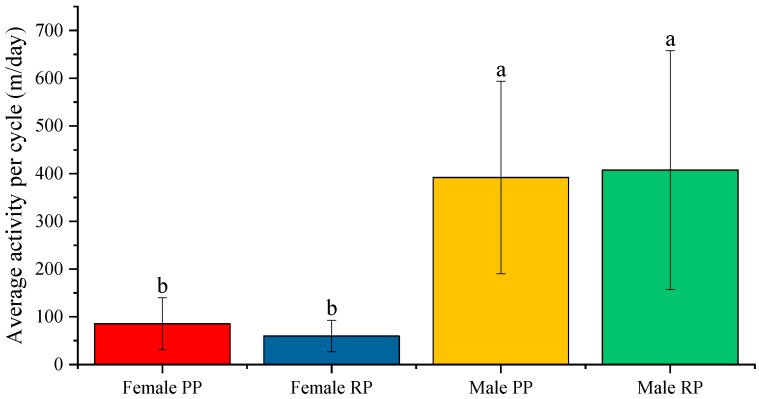
Average value of the mean activity per cycle (m/day) in the activity–rest circadian rhythm in male and female fish before and during the reproductive period. Different letters indicate statistically significant differences. The intervals above each bar represent the standard deviation.

**Figure 10 biology-12-00810-f010:**
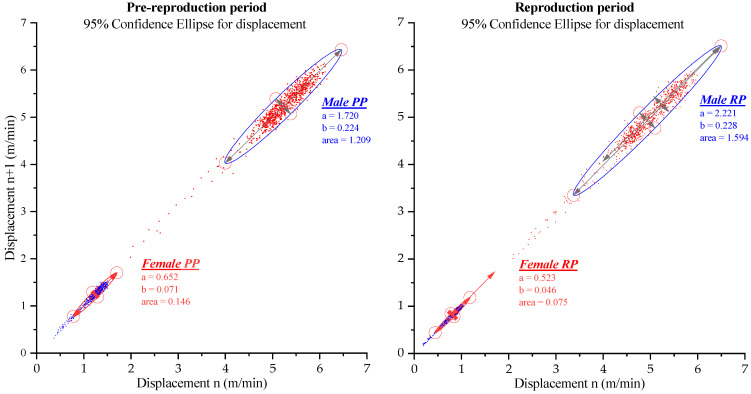
Distribution of activity in pre-reproductive periods and reproduction in Poincare diagram. The área of the 95% confidence ellipse, where a and b are the lengths of the major and minor axes of the ellipse, show that males travel greater distances in both periods.

**Figure 11 biology-12-00810-f011:**
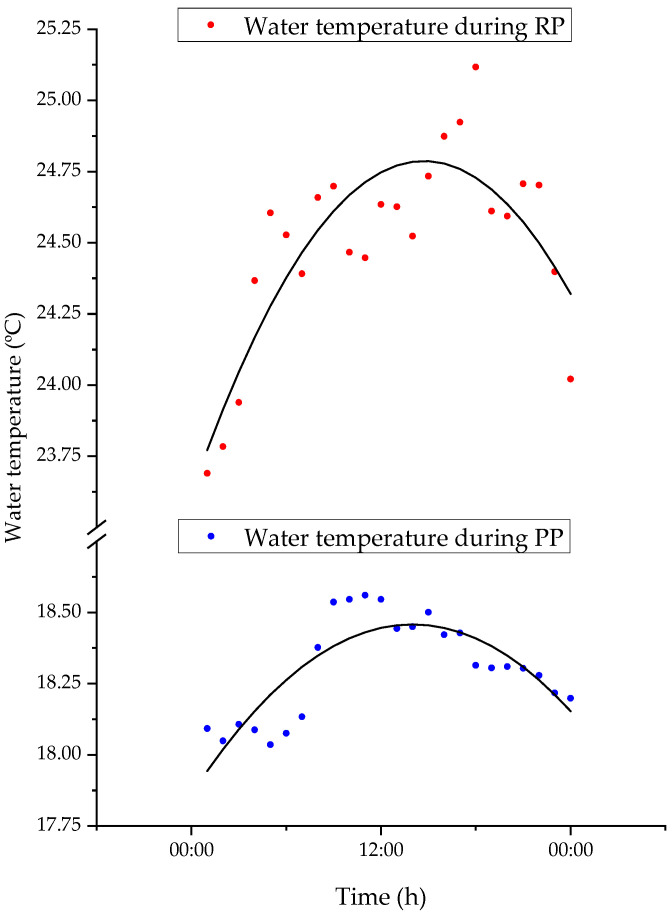
Average water temperature recorded every hour for 24 h during PP (bleu dots) and RP (red dots). The black lines represent Chebyshev polynomial fit.

**Figure 12 biology-12-00810-f012:**
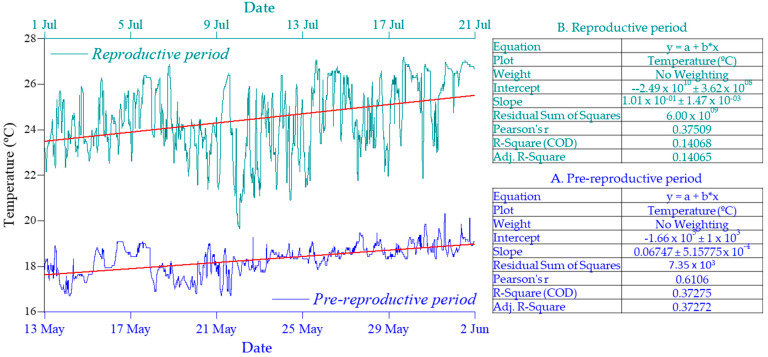
The trendline and Pearson correlation coefficient between water temperature and successive days of recording during PP (**A**) and RP (**B**).

**Figure 13 biology-12-00810-f013:**
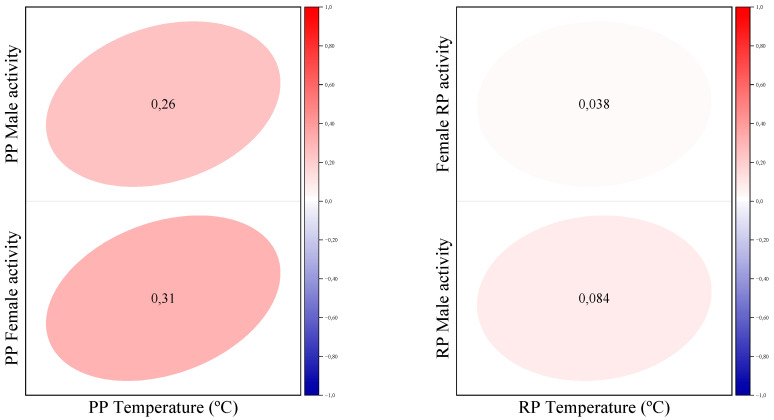
Pearson correlation coefficient between water temperature and fish activity during PP and RP.

## Data Availability

Not applicable.

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
