# Peer review of "Activity–Rest Circadian Rhythm of the Pearly Razorfish in Its Natural Habitat, before and during Its Mating"

_biology, 2023, doi:10.3390/biology12060810_

Round 1

Reviewer 1 Report

GENERAL COMMENTS

The paper of Mourad Akaarir and colleagues observe the circadian activity-rest rhythm of Pearl Razorfish in its natural habitat, before and during mating according to sex (male and female).

This paper falls within the scope of Biology.

The subject of the manuscript is very specific and I'm not sure of the interest it might arouse in most readers. Therefore, the authors should explain the importance of knowing these aspects about pearl razorfish and specify what are the implications and applications of these findings. For example, what does this information add to the knowledge of the biology of this fish and why do you think it is important? what are the direct and indirect effects on the environment and ecosystem?

Furthermore, the authors should better describe the study setup and the working principle of the monitoring system (system JSATS).

INTRODUCTION

Please review the formatting of the introduction (space between lines)

Please Insert a small introduction and description of this fish.

MATERIALS AND METHODS

L115 Please specify how you measured the length of the fish.

L116 Have you also recorded the body weight of the fish?

L134: Please specify the instrument used to measure the water temperature.

L137: How were daylight hours recorded?

L147: From this description it is not clear whether this product (system JSATS from Lotek Wireless) is applied to fish or only to the surrounding environment. Describe in more detail the methods of application and the method to install acoustic receivers.

-          It would be useful for the reader to insert images or diagrams that help to understand the system and its functioning.

-          Has the monitoring system been set up/calibrated before recording the data used for statistical analysis? If yes, for how long?

-          Has this system been validated for scientific purposes?

-          Also, did you observe whether there were influences of some hydrodynamic and therefore unstable conditions (such as the variation of water temperature during the day) on the observed data?

RESULTS

In Figure 1, correct the group names: Is Male PR, correct? or is it Male PP?

L246: Please remove the asterisks in the parenthesis.

L314: Please insert a space.

L321. The term "waking up time"  up is not accurate if you have previously argued that in this work it is not possible to distinguish sleep from rest and therefore we are only talking about rest. Maybe it's better to find a more appropriate and less misleading term.

DISCUSSION

L452: Have you speculated what these causes might be?

L491-494: "Finally, the differences observed in the start time of M10, which is later in the males, would also indicate the need for the male to defend his harem, starting this defence activity just before the females become active". Later o before?? In the results males start M10 earlier than females 7:20/8:14 vs 9:00.

In general, Don't you think that the stress due to the same observation/recording (handling the animals etc) of the data could have influenced the results? especially of the first days of observation?

REFERENCES

Reference number 1 and 2 are not in English. It would be better to report the English version.

Author Response

I am very grateful for the efforts of the two reviewers.

I am convinced that their commentaries and suggestions will contribute to greatly enhance the quality of our manuscript. Indeed, the entire text of our new version has been re-redacted. I also modified some figures according to your suggestions by removing those that were unnecessary and adding a few others that will help to clarify the report.

I hope that that these changes as well as the detailed answers to the questions raised after the first reading of our manuscript will satisfy the requirements for the publication of our report.

Yours sincerely

Reviewer 2 Report

The described study is valuable and interesting, because it concerns behavioral changes in the diurnal rhythm of free-living marine fish of both sexes depending on the breeding season.

The strong aspect of the work is a modern equipment that allows for precise recording of the behavior of the individual specimens of examined fish, and a very sophisticated analysis of the recorded parameters. Unfortunately, it was not fully used in the Discussion of the results.

The weak point is the imprecise description of the experimental setup and the interpretation of the obtained results. Detailed comments are provided below.

1’ Abstract is not fully understood if read separately, mainly due to non-standard abbreviations introduced without explanation.

2’ Introduction: According to the title, the article describes changes in the circadian rhythm of activity-rest of the pearly razorfish depending on the season (pre-reproductive and reproductive). This experimental approach has no direct connection with sleep, therefore the large fragments of the Introduction describing sleep, mainly in mammals (lines 73-92), are
redundant and should rather be removed. The same applies to the anatomy and function of the pineal organ and role of melatonin in fish (lines 59-72), because none of these aspects have been examined in the study.

3’ M & M are not described properly.

i - L:D and temperature conditions in both seasons are not described precisely; description of the diurnal changes in both parameters in experimental seasons should be very useful for further interpretation of the results obtained (see later);  

ii - It would be very useful (especially for readers not familiar with the tropical fish) to show photos of the tested species of fish (individuals of both sexes) and also to indicate the size (if not photos) of the sensors installed into the animals examined in the study;  

Iii - It is not clear whether the two groups of males and females tested in the two seasons were the same individuals or whether some of the fish with implanted sensors were tested in each of the two seasonal conditions (lines 126-128: The study comprises four groups…);  

iv - The water temperature is given as an average value with a significant difference in both seasons (approx. 6 degrees C), while the length of the day differed only by 15 minutes.
It is regrettable that daily temperature fluctuations were not shown instead of the average value - as it seems, during the entire recording period, i.e. 20 days.

The authors seem not to pay attention to this at all, while it is known that in fish the pineal gland responds not only to the photoperiod but also to the thermoperiod and it would be worth to be taken into consideration.

As far as the Discussion of the results is concerned, the question arises:

What is the seasonal time giver (Zeitgeber) that triggers reproductive activity in the studied fish – could it be the change in temperature that is the environmental signal much better expressed than a change in day length?

Discussion mainly raises the problem of the length of day, which differs between seasons by only 15 minutes – do you have any data indicating that a 15-minute reduction in the duration of night has such a profound effect on melatonin synthesis that it triggers reproductive behavior?

It is suggested to analyze and discuss these relationships instead of describing in details the differences in the personality of particular male individuals, which was neither recorded during the study nor analyzed as a parameter of the diurnal rhythm of behavior.  

Finally – Conclusions should not contain references to the literature, which is already presented in Discussion.

Author Response

Response to Reviewer 2 Comments

I am very grateful for the efforts of the two reviewers.

I am convinced that their commentaries and suggestions will contribute to greatly enhance the quality of our manuscript. Indeed, the entire text of our new version has been re-redacted. I also modified some figures according to your suggestions by removing those that were unnecessary and adding a few others that will help to clarify the report.

I hope that that these changes as well as the detailed answers to the questions raised after the first reading of our manuscript will satisfy the requirements for the publication of our report.

Yours sincerely

GENERAL COMMENTS

The described study is valuable and interesting, because it concerns behavioral changes in the diurnal rhythm of free-living marine fish of both sexes depending on the breeding season. The strong aspect of the work is a modern equipment that allows for precise recording of the behavior of the individual specimens of examined fish, and a very sophisticated analysis of the recorded parameters. Unfortunately, it was not fully used in the Discussion of the results.

We thank the reviewer all these positive comments that highlight our work.

The weak point is the imprecise description of the experimental setup and the interpretation of the obtained results. Detailed comments are provided below.

Answer: We agree with the reviewer that the experimental setup was not properly described in the previous version of the ms. According to the comments of reviewer 1, we have edited this section introducing more detail and a new figure (Fisgure 1SM) describing the configuration of the tracking system.

1’ Abstract is not fully understood if read separately, mainly due to non-standard abbreviations introduced without explanation.

Answer: Abbreviation’s have been described

2’ Introduction: According to the title, the article describes changes in the circadian rhythm of activity-rest of the pearly razorfish depending on the season (pre-reproductive and reproductive). This experimental approach has no direct connection with sleep, therefore the large fragments of the Introduction describing sleep, mainly in mammals (lines 73-92), are redundant and should rather be removed. The same applies to the anatomy and function of the pineal organ and role of melatonin in fish (lines 59-72), because none of these aspects have been examined in the study.

Answer: We agree with the reviewer, and we have reduced the introduction, particularly removing sleep-like section

3’ M & M are not described properly.

i - L:D and temperature conditions in both seasons are not described precisely; description of the diurnal changes in both parameters in experimental seasons should be very useful for further interpretation of the results obtained (see later);  

Answer: We have added temperature data to our manuscript, both in introduction, methods, results and discussion.

ii - It would be very useful (especially for readers not familiar with the tropical fish) to show photos of the tested species of fish (individuals of both sexes) and also to indicate the size (if not photos) of the sensors installed into the animals examined in the study;  

Answer: A picture of a male and a female of Xyrichtys novacula has benn includes in Figure 1SM for non-familiar readers

Iii - It is not clear whether the two groups of males and females tested in the two seasons were the same individuals or whether some of the fish with implanted sensors were tested in each of the two seasonal conditions (lines 126-128: The study comprises four groups…);  

Answer: Analysis was focused in the individuals with data in both periods.

iv - The water temperature is given as an average value with a significant difference in both seasons (approx. 6 degrees C), while the length of the day differed only by 15 minutes. It is regrettable that daily temperature fluctuations were not shown instead of the average value - as it seems, during the entire recording period, i.e. 20 days.

Answer: we added a new graphic showing the oscillations in the Tº during the experiment.

The authors seem not to pay attention to this at all, while it is known that in fish the pineal gland responds not only to the photoperiod but also to the thermoperiod and it would be worth to be taken into consideration.

Answer: In fish, the pineal gland responds not only to the photoperiod but also to the thermoperiod. We have added temperature data to our manuscript.

As far as the Discussion of the results is concerned, the question arises:

What is the seasonal time giver (Zeitgeber) that triggers reproductive activity in the studied fish – could it be the change in temperature that is the environmental signal much better expressed than a change in day length? Discussion mainly raises the problem of the length of day, which differs between seasons by only 15 minutes – do you have any data indicating that a 15-minute reduction in the duration of night has such a profound effect on melatonin synthesis that it triggers reproductive behaviorIt is suggested to analyze and discuss these relationships instead of describing in details the differences in the personality of particular male individuals, which was neither recorded during the study nor analyzed as a parameter of the diurnal rhythm of behavior.  

Finally – Conclusions should not contain references to the literature, which is already presented in Discussion.

Answer: Removed

Reviewer 3 Report

In this MS, the authors evaluate the circadian rhythm of activity-rest of the Pearly razorfish, Xyrichtys novacula in its own habitat, before and during the reproduction season. I think the manuscript is interesting, however, some comments are given below and I'd like to recommend the authors to take in consideration.

  1. Abstract: some numerical value and statistic (p value) of the results could be included in the abstract to make it more interesting to potential readers. The novelty should be highlighted clearly.

2.       Graphical Abstract: should be added summarizing the findings of the study.

3.       Abbreviations: should be defined the first time they appear. When defined for the first time, the abbreviations should be added between parentheses after the written-out form. Please check the following examples (IV, IS, RA, M10 and L5).

4.       In the introduction: Should be enriched with other studies mainly the fact that the references 4 and 5 occupy the whole introduction.

5.       Experimental animals: please add the protocol code and date of approval.

6.       Add a statistic (a or b) in all figure legend.

7.       Please follow the instructions of the journal (Author Contributions, Funding, Conflicts of Interest..............)

8.       Figure 1: How many Actograms were used and imaged? 

Author Response

Response to Reviewer 3 Comments

I am very grateful for the efforts of the two reviewers.

I am convinced that their commentaries and suggestions will contribute to greatly enhance the quality of our manuscript. Indeed, the entire text of our new version has been re-redacted. I also modified some figures according to your suggestions by removing those that were unnecessary and adding a few others that will help to clarify the report.

I hope that that these changes as well as the detailed answers to the questions raised after the first reading of our manuscript will satisfy the requirements for the publication of our report.

Yours sincerely

 GENERAL COMMENTS

In this MS, the authors evaluate the circadian rhythm of activity-rest of the Pearly razorfish, Xyrichtys novacula in its own habitat, before and during the reproduction season. I think the ‎manuscript is interesting, however, some comments are given below, and I'd like to recommend the authors to take in consideration.

We thank the reviewer for all these positive comments.

  1. Abstract: some numerical value and statistic (p value) of the results could be included in the abstract to make it more interesting to potential readers. The novelty should be highlighted clearly.

Answer: we have added the p value to the abstract

  1. Graphical Abstract: should be added summarizing the findings of the study.
  2. Abbreviations: should be defined the first time they appear. When defined for the first time, the abbreviations should be added between parentheses after the written-out form. Please check the following examples (IV, IS, RA, M10 and L5),

Answer: We have added the definitions of abbreviations in the abstract

  1. In the introduction: Should be enriched with other studies mainly the fact that the references 4 and 5 occupy the whole introduction.

Answer: We have revised the introduction and added a new paragraph with relevant references.

  1. Experimental animals: please add the protocol code and date of approval.

Answer: We have added the following protocol to the manuscript

This study "Activity-rest circadian rhythm of the Pearly razorfish in its natural habitat, before and during its reproduction" was approved by the Ethical Committee for Animal Experimentation of the University of the Balearic Islands (reference of the protocol CEEA 107/01/19, date February 20th, 2019) and was authorized by the Conselleria d'Agricultura, Pesca i Aliemntació of the Government of the Balearic Islands (reference of the authorization, 2019/20/AEXP). The Conselleria d'Agricultura, Pesca i Aliementació of the Government of the Balearic Islands also granted permission to capture, manipulate and release the animals in the Marine Protected Area of Palma Bay (reference of the authorization, 2085, February 11th, 2019.  The tagging protocol followed the guidelines provided by the Spanish Government (RD 53/2013).

  1. Add a statistic (a or b) in all figure legend.

Answer: We have added to the legend the letters a and b indicative of statistical significance.

  1. Please follow the instructions of the journal (Author Contributions, Funding, Conflicts of Interest..............)

Answer: We have added Funding and conficts of interest.

  1. Figure 1: How many Actograms were used and imaged? 

Answer: We have used one actogram and one image.

Round 2

Reviewer 1 Report

In the new version of the manuscript, I don't see some corrections that I suggested to the authors during the first round. Therefore, I would ask the authors to review my previous suggestions more carefully and try to correct where possible.

Author Response

Please find attached the revised manuscript

Reviewer 2 Report

The new version of the text is much better, although it has become a bit too long – especially the Discussion is too extended.  The lines 643-645 and 667-668 contain the expressions hard to understand (language mistakes?).

Figs. 7 and 10 are repeated twice

Newly added Fig. 11 is of greatest interest! However, it is recommended (suggested) that both lines (blue and red) should be drawn in the same figure and in the same range of values on vertical axis. Then the temperature differences between the two periods will be clearly (much better) visible.

Chapter „Conclusions” contains actually a Summary and that’s what it should be titled

Author Response

Please find attached our reply

Reviewer 3 Report

The authors followed the most of the comments suggested by the reviewers and therefore the MS has been improved accordingly.

Author Response

Please find attached our reply
